# The Impact of SGLT2 Inhibitors in the Heart and Kidneys Regardless of Diabetes Status

**DOI:** 10.3390/ijms241814243

**Published:** 2023-09-18

**Authors:** Jennifer Matthews, Lakshini Herat, Markus P. Schlaich, Vance Matthews

**Affiliations:** 1Royal Perth Hospital Unit, Dobney Hypertension Centre, School of Biomedical Sciences, University of Western Australia, Crawley, WA 6009, Australia; jen.matthews@uwa.edu.au (J.M.); lakshini.weerasekera@uwa.edu.au (L.H.); 2Royal Perth Hospital Unit, Dobney Hypertension Centre, School of Medicine, University of Western Australia, Crawley, WA 6009, Australia; markus.schlaich@uwa.edu.au; 3Department of Cardiology and Department of Nephrology, Royal Perth Hospital, Perth, WA 6000, Australia

**Keywords:** diabetes, cardiovascular, renal, SGLT1, SGLT2, therapy

## Abstract

Chronic Kidney Disease (CKD) and Cardiovascular Disease (CVD) are two devastating diseases that may occur in nondiabetics or individuals with diabetes and, when combined, it is referred to as cardiorenal disease. The impact of cardiorenal disease on society, the economy and the healthcare system is enormous. Although there are numerous therapies for cardiorenal disease, one therapy showing a great deal of promise is sodium-dependent glucose cotransporter 2 (SGLT2) inhibitors. The SGLT family member, SGLT2, is often implicated in the pathogenesis of a range of diseases, and the dysregulation of the activity of SGLT2 markedly effects the transport of glucose and sodium across the luminal membrane of renal cells. Inhibitors of SGLT2 were developed based on the antidiabetic action initiated by inhibiting renal glucose reabsorption, thereby increasing glucosuria. Of great medical significance, large-scale clinical trials utilizing a range of SGLT2 inhibitors have demonstrated both metabolic and biochemical benefits via numerous novel mechanisms, such as sympathoinhibition, which will be discussed in this review. In summary, SGLT2 inhibitors clearly exert cardio-renal protection in people with and without diabetes in both preclinical and clinical settings. This exciting class of inhibitors improve hyperglycemia, high blood pressure, hyperlipidemia and diabetic retinopathy via multiple mechanisms, of which many are yet to be elucidated.

## 1. Introduction

Both Chronic Kidney Disease (CKD) and Cardiovascular Disease (CVD) are two of the most prevalent diseases globally. Chronic Kidney Disease is a progressive condition that affects over 10% (over 800 million) of the global population [1] and may ultimately lead to the final stage of kidney disease, known as end-stage kidney disease. However, according to the World Health Organization, CVD is the leading cause of death globally.

An even more disturbing phenomenon occurring now is the combination of cardiac and renal dysfunction, known as cardiorenal disease. In fact, it is now known that many individuals dealing with CKD do not actually reach the stage of dialysis, because they die of heart disease [2]. This is the case because CKD is multifactorial and could be related to a variety of reasons, including impaired coronary flow reserve (ratio of the maximal or hyperemic flow down a coronary vessel to the resting flow), reduced aortic compliance (the ability of the arterial wall to distend and increase volume with increasing transmural pressure), increase in the level of angiotensin II and changes in the concentrations of essential vitamins like potassium, calcium and magnesium, as well as fibrosis in the hearts of patients on dialysis [3].

Cardiovascular Diseases (CVD) are the leading cause of disease burden throughout the world, with prevalent cases of CVD doubling from 271 million people in 1990 to 523 million people in 2019 and the number of CVD deaths increasing from 12.1 million in 1990 to 18.6 million in 2019, and it is more fatal in men than women [4]. Treatment of CVD is complex, as there are a multitude of underlying causes of cardiovascular death, such as ischemic heart disease, stroke, hypertensive heart disease, cardiomyopathy/myocarditis, atherosclerosis, aortic aneurysm and peripheral artery disease. Alarmingly, modifiable risk factors such as high systolic blood pressure, high fasting plasma glucose, high LDL cholesterol, high BMI and poor diet choices are also on the rise [4]. In a study including 197 countries, the total costs of heart failure alone in 2012 was USD $108 billion [5].

Diabetes is one of the leading causes of death and disability worldwide, with 529 million people living with this condition as of 2021, resulting in health expenditures of USD $966 billion globally, and they are forecasted to reach more than USD $1054 billion by 2045. Type 2 diabetes makes up 96% of all diabetes cases worldwide, and it is equally prevalent in both males and females [6]. Diabetic Kidney Disease (DKD) develops in approximately 40% of patients with type 2 diabetes (T2D) and 30% of patients with type 1 diabetes (T1D), and it is the leading cause of chronic kidney disease (CKD) and end-stage renal disease [7,8]. The mortality risk associated with DKD has increased by 31.1%, and it increases with worsening disease severity [9]. It is reported that DKD affects males and females equally, and it rarely develops before 10 years of duration of T1D [10]. The humanistic, societal and economic impact of DKD is enormous. It places a significant burden on the health care system and seriously affects the physical health and quality of life of patients [11]. Globally, in 2019, there were 2.6 million cases of CKD due to diabetes mellitus [12].

At present, there are a multitude of therapies that are used for cardio-renal disease. Some of these include pharmacological agents such as diuretics, vasodilators, angiotensin-converting enzyme (ACE) inhibitors or angiotensin receptor blockers (ARBs) [13]. In fact, it has been shown that in an Italian diabetic population affected by both DKD and diabetic retinopathy, multifactorial intervention using a combination of angiotensin-converting enzyme inhibitors and angiotensin II receptor blockers resulted in improved outcomes [14]. Although standard treatments have been found to slow the progression of CKD, they do not halt the disease. Therefore, alternative treatments may be required, such as the non-steroidal mineralocorticoid receptor antagonist finerenone, which is used to treat DKD with albuminuria [15]. Other emerging therapies that are currently undergoing clinical trials include endothelin receptor-A antagonists, complement inhibition, Janus kinase (JAK) inhibition, chemokine inhibition, renal denervation and, of course, the topic of this review, sodium glucose cotransporter 2 (SGLT2) inhibitors.

## 2. What Are Sodium Glucose Cotransporters?

The entry of glucose into cells is regulated by facilitative glucose transporters (GLUTs) and sodium-dependent glucose cotransporters (SGLTs). Of the SGLT family, SGLT1 and SGLT2 are frequently investigated in a range of disease settings [16], as they play key roles in the transport of glucose and sodium across the brush-border membrane of intestinal and renal cells [17]. Although SGLT1 is less researched than SGLT2, it is more widely expressed throughout the body and is found predominantly in the small intestine [18,19,20] (Figure 1; Table 1), and it only accounts for 5–10% of the glucose reabsorption in the kidneys. Our studies have shown that heightened Sympathetic Nervous System (SNS) activity upregulates this SGLT1 expression, and therefore, inhibition of this protein could also be beneficial in treating cardiometabolic disorders [21]. SGLT2 is a high-capacity, low-affinity glucose cotransporter, mainly found in the S1 and S2 segments of the renal convoluted proximal tubules (Figure 1; Table 1), and it is required for the reabsorption of a majority of the glucose (~90–95%) filtered by the kidneys [22,23].

SGLT2 inhibitors (SGLT2i’s) have been developed based on the antidiabetic action initiated by inhibiting renal glucose reabsorption, thereby increasing urinary glucose excretion [24] (Table 2). As demonstrated by several large-scale clinical trials [25,26,27], SGLT2i’s are now recognized to be capable of altering a range of metabolic and biochemical parameters via novel mechanisms, which are discussed in this review, thereby exerting cardio-renal protection in individuals with and without diabetes [28,29].

One critical requirement before SGLT2i’s can be prescribed for the treatment of CKD is a functional glomerular filtration rate (GFR). The glomerular filtration rate is considered the optimal way to measure kidney function. In a healthy kidney, the glomerular filtration rate is 120 mL/min/1.73 m^2^, but as CKD progresses, it may decline to 60 mL/min/1.73 m^2^ or less. Interestingly, studies have shown that SGLT2i’s are safe and beneficial to use in patients with a GFR above 20 mL/min/1.73 m^2^ and can actually be used in subjects that have a GFR of less than 20 mL/min/1.73 m^2^, as long as they are tolerating it well and are not on dialysis [30]. However, this topic of discussion is controversial, and a number of other specific aspects need to be addressed. Currently, SGLT2 inhibitors can be used in CKD patients up until starting dialysis due to their nephro- and cardioprotective effects. It is also believed that SGLT2 inhibitors may be used safely in hemodialysis and peritoneal dialysis patients, especially if they have additional heart failure.

**Table 1 ijms-24-14243-t001:** Areas of expression of SGLT1 and SGLT2 in the human body.

Location	SGLT1	SGLT2
Small Intestine	Apical membrane, K and L cells [19].	Not expressed.
Eye	Retina [20].	Retina, cornea and lens [20].
Kidney	Section 3 of the proximal tubule. [21].	Section 1 and 2 of the proximal tubules [22].
Pancreas	Pancreatic alpha cells [18].	Not expressed.
Liver	Biliary duct cells [19].	Not expressed.
Heart	Capillaries [19].	Not expressed.

**Table 2 ijms-24-14243-t002:** Summary of key findings from the characterization of Canagliflozin (CANA), Dapagliflozin (DAPA), Ipragliflozin, Empagliflozin (EMPA), Tofogliflozin and Luseogliflozin [31,32,33].

Study Parameters	Key Findings
Pharmacokinetic properties	Longest plasma half-life: Canagliflozin.
	Longest half-life in the kidney: Dapagliflozin.
	Highest distribution in the kidney: Ipragliflozin.
	Drug distribution in the kidney suggested to be dependent on chemical structure.
Pharmacodynamic properties	All SGLT2i’s increased urinary glucose excretion in a dose-dependent manner.
	Long-acting SGLT2i’s exhibited persistent action, even 18 h post dose.
	Close correlation between the duration of action, plasma drug concentration, drug distribution and kidney retention.
Pharmacologic properties	Significant reductions in blood glucose and plasma insulin with all SGLT2i’s.
	Significant improvement in glucose tolerance with all SGLT2i’s.
	Long-acting SGLT2i’s exert stronger anti-hyperglycaemic effects through persistent urine glucose excretion.
	Intermediate-acting SGLT2i’s may provide better glycaemic control when administered twice daily.
Anti-diabetic effects	All SGLT2i’s significantly improved hyperglycaemia and hyperinsulinemia.
	All SGLT2i’s significantly increased pancreatic insulin content by prevention of pancreatic exhaustion.
	Long-acting SGLT2i’s exert favourable glycaemic control over 24 h and may have slightly enhanced antidiabetic effects compared with intermediate-acting SGLT2i’s.
Effects on diabetic complications	All SGLT2i’s exhibited significant improvements/trends in obesity parameters (e.g., body and visceral fat weights, lipid metabolism markers), proinflammatory cytokines and endothelial dysfunction markers.
	All SGLT2i’s significantly decreased or showed a decreasing trend in steatohepatitis parameters (e.g., liver weight, plasma levels of liver enzymes) and renal parameters (e.g., creatinine clearance, renal tubular injury markers).
	Long-acting SGLT2i’s (0.3 mg/kg) demonstrated slight superiority in comparison with intermediate-acting SGLT2i’s (3 mg/kg) on several parameters (e.g., daily blood glucose control, visceral fat weight).

Blue colouration is related to pharmacological effects. Green colouration is related to anti-diabetic effects. Pink colouration is related to effects on diabetic complications.

## 3. Use of Selective SGLT2 Inhibition as an Antidiabetic Therapy

There are many studies that have shown the beneficial cardiovascular and renal effects that SGLT2i’s can have in both T1D and T2D. In patients with T2D, SGLT2i treatments are now being considered as the first line of therapy because of their metabolic and cardio-renal benefits. As just mentioned, SGLT2i’s provide benefits by promoting the excretion of glucose in the urine, therefore assisting with a reduction in hyperglycemia and subsequently a decrease in weight. SGLT2 inhibitors also promote both uric acid reduction [34,35] and stimulation of erythropoiesis, which may have cardiovascular and renal effects [34]. However, aside from the metabolic benefits of SGLT2i’s, there are numerous other beneficial protective mechanisms, as outlined below.

### 3.1. Preclinical Studies

#### 3.1.1. Blood Pressure Reduction

One common complication of patients with diabetes is hypertension, and although the anti-hypertensive therapies have improved over the years, there is definitely still room for improvement. Although SGLT2i’s are not predominantly prescribed as hypertensive medications, research has now shown that SGLT2i’s have overwhelming beneficial effects in patients with both hypertension and diabetes. Mechanisms that may contribute to their anti-hypertensive action are their mild natriuresis, osmotic diuresis and weight-loss effects in patients with diabetes [36]. In human studies, both Empagliflozin (EMPA) [37] and Dapagliflozin (DAPA) [38] were associated with significant reductions in blood pressure in patients with diabetes compared with placebo. We have also demonstrated in our murine studies that SGLT2i’s may promote sympathoinhibition in the kidneys and heart in diabetic mice, and this may be an underlying mechanism for blood pressure reduction [39].

#### 3.1.2. Improved Digestive Health

The intestinal microbiota is aggravated as diabetes progresses and, during the development of DKD, there is an increased imbalance in the gut microbiota. The SGLT2i DAPA has been beneficial in lowering the level of dysbacteriosis and bile acids [40], altering the microbiota composition [41] and reducing succinate levels (pathogenic factor in diabetic retinopathy) [42], while the inhibitor EMPA reduces the lipopolysaccharide (LPS)-producing bacteria and increases the short-chain fatty acid (SCFA)-producing bacteria [43]. Additional inhibitors such as Luseogliflozin have also been found to be beneficial in altering the microbiota. When mice were given the SGLT2 inhibitor Luseogliflozin, there was a significant increase in the abundance of the species *Syntrophothermus lipocalidus*, *Syntrophomonadaceae* and *Anaerotignum*, which are all involved in the biosynthesis of important SCFAs such as acetic acid, propionic acid and butyric acid [44].

#### 3.1.3. Diabetic Retinopathy

Diabetic retinopathy is a common complication associated with diabetes, and the SGLT2i’s EMPA [45] and DAPA [46] have improved diabetic retinopathy in T2D db/db mice. Empagliflozin has also been shown to mitigate ocular edema and microaneurysms in the retina as well as inhibit the mammalian target of raptomycin activation. One of the characteristics of diabetic retinopathy in db/db mice is an increase in acellular capillary numbers [46], and DAPA has been shown to produce a substantial decrease in the acellular capillary numbers compared with placebo. Aside from this, inflammation is another main factor associated with diabetic retinopathy. According to these studies, both EMPA and DAPA have been shown to downregulate inflammatory and angiogenic factors, such as Tumor Necrosis Factor-α (TNF-α) [45,46], Interleukin 1-β (IL1-β) [46], Interleukin 6 (IL-6) [45], Vascular Cell Adhesion Molecule 1 (VCAM-1) [45] and Vascular Endothelial Growth Factor (VEGF) [45] in the retina.

#### 3.1.4. Kidney Health

There have been numerous preclinical studies conducted which show the renoprotective effects of SGLT2i’s. Dapagliflozin treatment in db/db mice halted the progressive increases in albumineria and glomerulosclerosis [47], while in the Otsuka Long–Evans Tokushima Fatty (OLETF) T2D rat model, DAPA was shown to reverse renal oxidative stress markers as well as attenuate inflammatory cell infiltration, mesangial widening, interstitial fibrosis and total collagen content [48]. Luseogliflozin treatment in a T2D nephropathy rat model has been shown to prevent the fall in GFR and reduce the degree of glomerular injury, renal fibrosis and tubular necrosis compared with those on vehicle or insulin alone [49]. Empagliflozin treatment has also been found to ameliorate albuminuria and glomerular injury in db/db mice [50].

#### 3.1.5. Cardiovascular Benefits

There have been many preclinical animal studies showing the cardioprotective effects of SGLT2i’s in T2D. When EMPA was administered to db/db mice over a 10-week period, it was shown to significantly ameliorate cardiac interstitial fibrosis, pericoronary arterial thickening, cardiac macrophage infiltration and the impairment of vascular dilation [50]. Further discussion of SGLT2i-mediated cardiovascular benefits is presented later in the review.

#### 3.1.6. Improved Cognitive Function in T2D

One of the most underrecognized but life-changing effects of T2D is the decline in cognitive function. Therefore, the question as to whether SGLT2i’s reduce the cognitive impairment associated with T2D has been investigated in numerous animal studies. Empagliflozin has been shown to significantly prevent the impairment of cognitive function in T2D db/db mice due to its ability to attenuate cerebral oxidative stress, as well as increase cerebral Brain-Derived Neurotrophic Factor (BDNF) levels [50].

### 3.2. Human Clinical Trials 

There are a multitude of clinical trials that have been conducted utilizing the SGLT2 inhibitors EMPA, DAPA, Canagliflozin (CANA) and Ertugliflozin, and they are outlined below (Table 3).

#### 3.2.1. Empagliflozin

The groundbreaking clinical trial assessing the effectiveness of the SGLT2 inhibitor EMPA in preventing Cardiovascular Disease in people with diabetes is the Empagliflozin Cardiovascular Outcome Event Trial in type 2 diabetes Mellitus Patients–Removing Excess Glucose (EMPA-REG) [25,51]. This trial demonstrated that when patients who had T2D as well as a high risk of cardiovascular events were treated with EMPA, mortality (3.7% vs. 5.9%) and hospitalizations (2.7% vs. 4.1%) due to heart failure were reduced compared with placebo [25]. Empagliflozin also promoted a slowing down of the progression of kidney disease (12.7% vs. 18.8%), as well as a significantly lower risk of clinically relevant renal events, including renal replacement therapy (0.3% vs. 0.6%) versus placebo controls [51].

#### 3.2.2. Dapagliflozin

When it comes to cardiovascular and kidney disease, there are two main clinical trials evaluating the effectiveness of the SGLT2 inhibitor DAPA in people with diabetes. One major cause of Cardiovascular Disease is atherosclerosis, which may be caused by both traditional and nontraditional risk factors [4] (Figure 2). The Dapagliflozin Effect on Cardiovascular Events–Thrombolysis in Myocardial Infarction 58 (DELCARE-TIMI 58) trial [26] recruited people who had both diabetes and atherosclerotic Cardiovascular Disease, and although Dapagliflozin did not result in a higher or lower rate of Major Adverse Cardiovascular Events (MACE), it did lead to a lower rate of cardiovascular death or hospitalization for heart failure (4.9% vs. 5.8%) compared with placebo. In addition to this trial, the Dapagliflozin and Prevention of Adverse Outcomes in Chronic Kidney Disease (DAPA-CKD) study [57] found that not only did DAPA reduce the risk of a declining GFR level (5.2% vs. 9.3%), it decreased the risk of end-stage kidney disease (5.1% vs. 7.5%) and death from renal causes (<0.1% vs. 0.3%) versus controls.

#### 3.2.3. Canagliflozin

There are also two main clinical trials determining the effectiveness of the SGLT2 inhibitor CANA on Cardiovascular Disease in people with type 2 diabetes. The first is the Canagliflozin Cardiovascular Assessment Study (CANVAS) [27], where it was found that CANA not only reduced the risk of major adverse cardiovascular events in T2D patients at an increased risk of Cardiovascular Disease, it also offered renoprotective benefits, such as the regression of albuminuria levels (8.9% vs. 12.9%) and a decrease in the need for renal-replacement therapy or death (0.55% vs. 0.9%) when compared with placebo. In the second trial, known as the Canagliflozin and Renal Endpoints in Diabetes with Established Nephropathy Clinical Evaluation (CREDENCE) study [52], it was concluded that in patients who had both T2D and kidney disease, the risk of kidney failure and cardiovascular events was lower in the CANA group compared with the placebo group.

#### 3.2.4. Ertugliflozin

The final major SGLT2 inhibitor clinical trial is the Evaluation of Ertugliflozin efficacy and Safety Cardiovascular Outcomes (VERTIS) study [53], where the effect of Ertugliflozin in patients with both T2D and Cardiovascular Disease was investigated. It was found that not only did this inhibitor reduce the risk for first and total hospitalizations for heart failure, it also reduced the risk of death from heart failure/Cardiovascular Disease (8.1% vs. 9.1%) versus controls.

## 4. Use of Selective SGLT2 Inhibition as a Nondiabetic Therapy

Although SGLT2 inhibitors were originally utilized for the treatment of type 2 diabetes specifically, this drug class is now also being used in people without diabetes to assist in protection against cardiorenal disease (Table 3).

### 4.1. Preclinical Studies

#### 4.1.1. Modulation of Sympathetic Nervous System Activity

The SNS is a driver of hypertension, hyperglycemia and chronic kidney disease and, remarkably, our team and others have shown that SGLT2i’s may protect the kidneys and the heart through their sympathoinhibitory abilities [22,62]. In Apo E−/− mice, it was discovered that with the use of EMPA, norepinephrine (NE), the marker of heightened sympathetic activity was partially inhibited [63]. Our study with neurogenically hypertensive BPH/2J mice showed that DAPA lowered the NE and tyrosine hydroxylase (TH) levels in heart and kidneys [22].

Interestingly, the SNS performs diverse functions in different tissues. For instance, our team discovered that SGLT2 inhibition in the white adipose tissue (WAT) actually promoted sympathoexcitation and beiging [64]. In a cohort of BPH/2J mice, DAPA was administered via oral gavage, and the mice were found to have an increased level of tyrosine hydroxylase (TH) and norepinephrine expression in the WAT [64]. This is particularly exciting, as beiging is a process that typically occurs during fasting or exercise. During beiging, there is an upregulation of Uncoupling Protein 1 (UCP1), but there is also an increase in the molecules’ α-Aminoisobutyric acid, irisin and fibroblast growth factor 21 (FGF-21) [65].

#### 4.1.2. Reductions in Blood Pressure

Studies have highlighted that SGLT2 inhibition promotes remarkable reductions in blood pressure. In our neurogenically hypertensive BPH/2J mice, treatment with DAPA significantly reduced systolic and diastolic blood pressure, as well as mean arterial pressure (Figure 3). This blood pressure reduction also correlated with a remarkable sympathoinhibitory effect in the kidneys [22].

#### 4.1.3. Inflammation Control

One contributing pathogenic factor when it comes to both Chronic Kidney Disease and heart disease is that of inflammation. In Apo E−/− mice, EMPA, DAPA and CANA were all found to reduce inflammatory markers. Empagliflozin reduced IL-1β and IL-6 [63] and DAPA reduced IL-1β, Interleukin 18 (IL-18) and the inflammasome marker NOD-LRR and Pyrin domain containing protein 3 (NLRP3) [66], while CANA reduced VCAM-1 and the Monocyte chemotactic protein-1 (MCP-1) inflammatory protein while increasing the TIMP metallopeptidase inhibitor 1 (TIMP-1) [24]. In our neurogenically hypertensive BPH/2J mice, we found that DAPA treatment reduced inflammation in the heart by significantly reducing the inflammatory cytokine IL-6, while at the same time increasing the anti-inflammatory cytokine Interleukin 10 (IL-10) [22].

#### 4.1.4. Increases in Ketone Levels

More and more research is now showing that ketone bodies are an efficient substrate for the heart [67] and the kidneys [68]. Empagliflozin has been shown in multiple studies to increase ketone bodies in patients with chronic heart failure [69], as well as in both ZSF1 [70] and ApoE knockout mice [71]. Although ketoacidosis is a topic of concern when it comes to utilization of SGLT2i’s, it is of much greater concern for those with T1D as opposed to T2D. Although ketoacidosis can occur in T2D patients not on insulin, a human study has shown that when T2D patients were admitted to hospital with COVID-19, those who were administered SGLT2i therapy were at no greater risk of ketoacidosis than those not taking the inhibitors [72]. The subject of whether ketones increase or decrease SNS activity is still a controversy. In a preclinical study, it was found that the ketone body β-hydroxybutyrate suppresses SNS activity by antagonizing G-protein-coupled receptor 41 (GPR41) [73]. However, an alternative study involving rats showed that increased ketone body utilization did not suppress SNS activity and may stimulate it similarly to results seen with carbohydrates or fats [74]. In our future studies, we aim to assess whether ketone bodies suppress SNS and, consequently, how they may promote cardiorenal health.

#### 4.1.5. Improved Cardiovascular Health

Cardiomyopathy is a condition caused by inflammation in the cardiomyocytes, often through HFD consumption. In an in vitro study involving cultured rat H9c2 cardiomyocyte cells, treatment with the SGLT2 inhibitor DAPA attenuated hypertrophy, fibrosis and apoptosis. Furthermore, HFD-fed mice were administered DAPA, and this treatment improved the lipid profile as well as alleviated HFD-induced cardiac dysfunction and cardiac inflammation [75].

#### 4.1.6. Steatosis and Insulin Resistance

It is well known that SGLT2i’s lead to blood glucose reductions and weight loss in T2D. However, the influence of SGLT2 inhibition on high-fat diet (HFD)-induced obesity and insulin resistance is less well known. Preclinical studies on C57BL/6J HFD mice found that when given the SGLT2i’s EMPA [76,77] or CANA [78], not only did it increase the urinary excretion of glucose, it also increased weight loss and attenuated hepatic steatosis in the treatment groups compared with vehicle. In addition, EMPA suppressed the HFD-induced weight gain by enhancing fat utilization and browning and attenuated obesity-induced inflammation and insulin resistance [76].

### 4.2. Human Clinical Trials

#### 4.2.1. Empagliflozin

Empagliflozin is an SGLT2 inhibitor that has been approved for use in people with diabetes, but it is now being increasingly used in people who do not have diabetes but do have cardiovascular and/or renal disease. In the Empagliflozin Outcome Trial in Patients with Chronic Heart Failure and a Reduced Ejection Fraction (EMPEROR-REDUCED) [56], it was concluded that patients on EMPA had a 25% lower risk of cardiovascular death or hospitalization for heart failure (19.4% vs. 24.7%) than subjects on placebo, irrespective of whether the patients had diabetes or not. Another trial, known as the Empagliflozin Outcome Trial in Patients with Chronic Heart Failure with Preserved Ejection Fraction (EMPEROR-PRESERVED) [58], determined that EMPA treatment reduced the risk and severity of a broad range of inpatient and outpatient worsening heart failure events, including a reduction in hospitalization for heart failure (8.6% vs. 11.8%) and death from cardiovascular causes (7.3% vs. 8.2%) versus controls, therefore offering protective benefits. Aside from the EMPEROR trials, another clinical trial, known as Empagliflozin in Acute Myocardial Infarction (EMMY), investigated the effect of EMPA on CVD [59]. This trial assessed the benefits of utilizing EMPA in patients following acute myocardial infarction and found that EMPA significantly improved the echocardiographic functional and structural parameters. In addition, the EMMY clinical trial demonstrated that EMPA promoted a greater reduction in the N-terminal prohormone of brain natriuretic peptide (NT-proBNP), which is a heart failure marker. Finally, in order to ascertain the effect of EMPA on renal health, the Study of Heart and Kidney Protection with Empagliflozin (EMPA-KIDNEY) trial [60] was conducted, which showed a lower risk of progression of kidney disease or death from cardiovascular causes (13.1% vs. 16.9%) compared with placebo.

#### 4.2.2. Dapagliflozin

The second SGLT2 inhibitor that has been found to be beneficial in treating cardiovascular and renal disease in people without diabetes is DAPA. In the Dapagliflozin and Prevention of Adverse Outcomes in Heart Failure (DAPA-HF) [54] and the Dapagliflozin Evaluation to Improve the Lives of Patients with Preserved Ejection Fraction Heart Failure (DELIVER) [61] clinical trials, patients on DAPA had a lowered risk of worsening heart failure (16.3% vs. 21.2% and 11.8% vs. 14.5%, respectively) or death from cardiovascular causes (9.6% vs. 11.5% and 7.4% vs. 8.3%, respectively) compared with placebo. To ascertain the effect DAPA has on renal health, the DAPA-CKD [57] and the Effects of the SGLT2 inhibitor Dapagliflozin on proteinuria in nondiabetic patients with Chronic Kidney Disease (DIAMOND) [55] trials were conducted. The DAPA-CKD trial showed a significant reduction in the risk of death from renal or cardiovascular causes (9.2% vs. 14.5%) compared with placebos. Meanwhile, the DIAMOND trial showed that while DAPA treatment did not affect proteinuria in patients with CKD, it reduced GFR levels compared with placebo. While reduced GFR is a sign of progressive kidney disease, there is a phenomenon that occurs with SGLT2i’s known as the estimated GFR (eGFR) acute dip, which is an acute reversible reduction in GFR. Of significance, the EMPA-REG [51], VERTIS-CV [53] and the CREDENCE [52] clinical trials have all confirmed that the dip in eGFR is not associated with progressive loss of long-term kidney function or acute kidney injury [79].

#### 4.2.3. Ipragliflozin

Although not one of the main SGLT2i’s, Ipragliflozin may also offer clinical potential, as demonstrated by a recent case study [80]. In this study, an 83-year-old man with chronic heart failure and T2D was hospitalized four times over 5 years, but with the use of Ipragliflozin, he displayed reduced cardiac sympathetic nerve activity and was not hospitalized for 2 years afterwards. The improved health of the patient may be due to the reduction in observed cardiac sympathetic nerve hyperactivity, and this finding warrants further investigation of this inhibitor [80].

## 5. Discussion

### 5.1. Are Dual SGLT1/2 Inhibitors More Effective Than Sole SGLT2 Inhibitors?

One of the burning questions at present is whether utilizing a dual SGLT1/2 inhibitor like Sotagliflozin (SOTA) may be more beneficial than sole SGLT2i’s such as EMPA, CANA or DAPA when it comes to the treatment of cardiorenal disease in both patients with and without diabetes. As dual SGLT1/2 inhibition is a relatively new pharmacological therapy compared with sole SGLT2 inhibitors, the breadth of research findings is limited.

A study conducted on nondiabetic C57BL/6J mice found that treatment with SOTA attenuated cardiac hypertrophy and histological markers of cardiac fibrosis which were induced by the transverse aortic constriction (TAC) procedure [81].

Two clinical trials utilizing SOTA are the Effect of Sotagliflozin on Cardiovascular Events in Patients with Type 2 Diabetes Post Worsening Heart Failure (SOLOIST-WHF) trial [82] and the Effect of Sotagliflozin on Cardiovascular and Renal Events in Participants with Type 2 Diabetes and Renal Impairment Who Are At Cardiovascular Risk (SCORED) trial [83]. Both the SOLOIST-WHF trial and the SCORED trial showed that in patients with T2D and CKD, the rate of hospitalization and urgent care visits for heart failure was reduced when patients were administered SOTA compared with placebo. A new clinical trial, called the Sotagliflozin in Heart Failure with Preserved Ejection Fraction Patients (SOTA-P-CARDIA) [84], is currently underway to investigate the SOTA-mediated cardiovascular effects and mechanisms of action in patients with heart failure with preserved ejection fraction (HFpEF) but without diabetes.

### 5.2. Interesting Avenues for SGLT2i Therapy for the Treatment of T1D

To date, SGLT2i’s have not been clinically approved for the sole treatment of T1D due to the concerns surrounding hypoglycemia and ketoacidosis. However, it is a field of research that is gaining momentum, both in the context of preclinical and clinical pilot studies.

#### 5.2.1. Animal Studies Utilizing SGLT2i’s as a Treatment for T1D

Our team has conducted a great deal of research utilizing our T1D Akimba mouse model. We have shown that SGLT2i’s may be a potential therapeutic for not just T2D but also T1D. When our Akimba mice were treated with the SGLT2i’s DAPA [85], CANA [86] and EMPA [86], metabolic parameters such as fasting blood glucose levels, polydipsia (excessive thirst) and weight management were all improved. Aside from the metabolic advantages in our T1D mice, we determined that DAPA and EMPA conferred beneficial effects on (i) digestive health [42] by significantly increasing the beneficial short-chain fatty acid butyric acid and (ii) diabetic retinopathy by reducing microvascular lesions [85,86,87] and reducing the pathogenic factor succinate [42].

Studies have shown that SGLT2i’s promote an upregulation of the family member SGLT1 in the kidneys (Figure 4) and also cause a reduction in kidney size and an improvement in renal histology [88]. The compensatory upregulation of SGLT1 with SGLT2i’s warrants the use of dual SGLT1/2 inhibitors such as SOTA.

While we have studied the effects of SGLT2i’s in our Akita and Akimba mice with regards to kidney health, our future studies aim to determine how these inhibitors may also improve cardiovascular health in our T1D Akita mice, as this strain is known to manifest diabetic cardiomyopathy with ageing [89].

**Figure 4 ijms-24-14243-f004:**
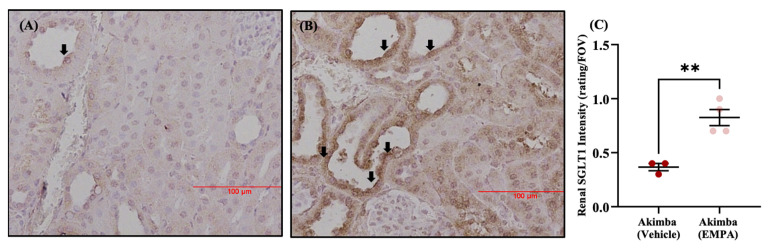
SGLT2 inhibition results in increased luminal SGLT1 expression in renal tissue of diabetic Akimba mice: Representative SGLT1 immunohistochemistry images of renal tissue of mice treated with (**A**) vehicle or (**B**) Empagliflozin (EMPA) via drinking water (25 mg/kg/day) for 8 weeks. (**C**) Quantitation of SGLT1 intensity; *n* = 3–4 mice/group; mean + SEM. ** *p* = 0.005. Black arrow = luminal SGLT1 expression. Intensity: 0 = absent–3 = high intensity; FOV, field of view. Figure taken from [90].

#### 5.2.2. Human Pilot Studies/Clinical Trials Utilizing SGLT2i’s as an Add-On to Insulin for Patients with Type 1 Diabetes

Due to the difficulty in managing insulin in T1D patients, there are many clinical trials and pilot studies that have investigated the effect of utilizing the three main SGLT2 inhibitors, CANA [91], DAPA [92,93] or EMPA [94], as adjunctive therapies alongside insulin. While the inhibitors promoted ketoacidosis (KA), it was indicated that it was related to inadequately controlled insulin. Proper monitoring of glucose and ketone levels, as well as titration of the inhibitor/insulin, may be able to control the incidence of ketoacidosis. The Empagliflozin as Adjunctive to Insulin Therapy (EASE) trial determined that a dose of 2.5 mg/day of EMPA (phase 3) as opposed to a dose of 10 mg/day of EMPA (phase 2) helped to reduce the incidence of KA and, therefore, the lower dose may be a viable treatment option.

All clinical trials, including the Study of Effects of Canagliflozin as Add-on Therapy to Insulin in the Treatment of Participants With Type 1 Diabetes Mellitus (T1DM) [91] trial, the Dapagliflozin Evaluation in Patients With Inadequately Controlled Type 1 Diabetes (DEPICT-1) [92] and Efficacy and Safety of Dapagliflozin in Patients With Inadequately Controlled Type 1 Diabetes (DEPICT-2) [93] trials, as well as the EASE-2/3 [94] Empagliflozin trials, showed that SGLT2i’s promoted reductions in HbA1c, body weight and insulin requirements. Additionally, the aforementioned clinical trials most importantly improved glycemic control [95,96] after treatment, without producing any hypoglycemia.

### 5.3. Use of Dual SGLT 1/2 Inhibitors in T1D

As highlighted above, there are many benefits to utilizing SGLT2i’s for the treatment of T1D. This review highlighted the promising emerging studies that have been conducted to show that SOTA is beneficial for T2D [97]. A question that remains is whether the dual SGLT1/2 inhibitor, SOTA, may be beneficial for T1D and even surpass the benefits conferred by sole SGLT2 inhibitors. In our recent study utilizing our T1D Akimba mice [90], we concluded that SOTA not only significantly decreased fasting blood glucose levels but also promoted healthy weight gain compared with vehicle counterparts. Aside from this, SOTA also improved diabetes-associated polydipsia. As mentioned previously, overactivation of the SNS is strongly associated with diabetes as well as cardiorenal disease and, therefore, understanding the effects that SOTA has on SNS activation is of critical importance. We found that SOTA therapy resulted in a reduction in the main neurotransmitter of the SNS, NE. This phenomenon is otherwise known as sympathoinhibition.

## 6. Conclusions

We clearly demonstrated that SGLT2 inhibitors have enormous potential in improving renal and cardiovascular outcomes in patients with or without diabetes (Figure 5). Although most research has been conducted on T2D, the use of SGLT2i’s in the treatment of T1D has also been found to be beneficial and definitely warrants further research.

## Figures and Tables

**Figure 1 ijms-24-14243-f001:**
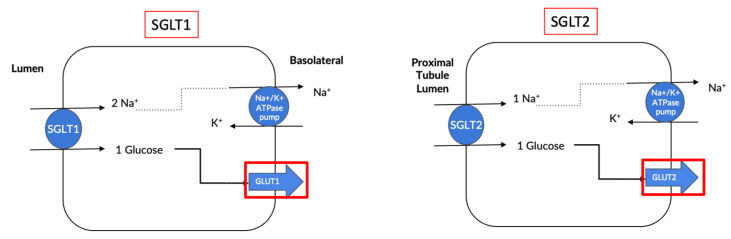
SGLT1 and SGLT2 sodium and glucose transport in intestinal lumen (SGLT1) and proximal tubule lumen in the kidneys (SGLT2).

**Figure 2 ijms-24-14243-f002:**
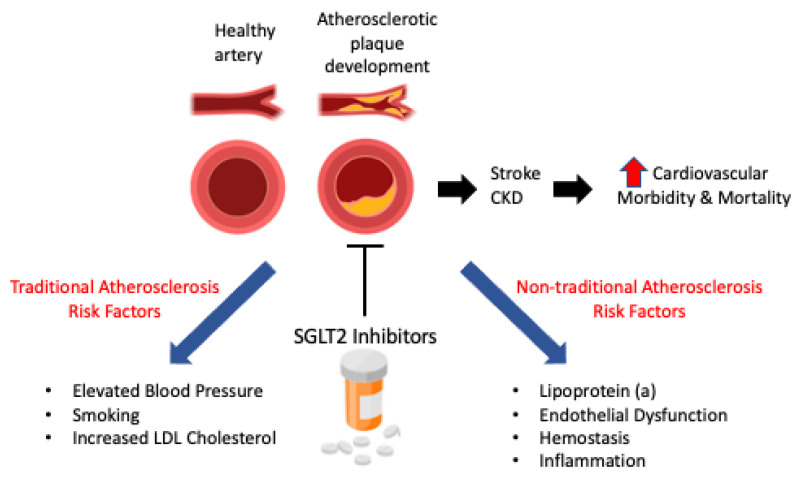
Traditional and nontraditional risk factors for atherosclerosis. Red arrow indicates an increase in cardiovascular morbidity and mortality.

**Figure 3 ijms-24-14243-f003:**
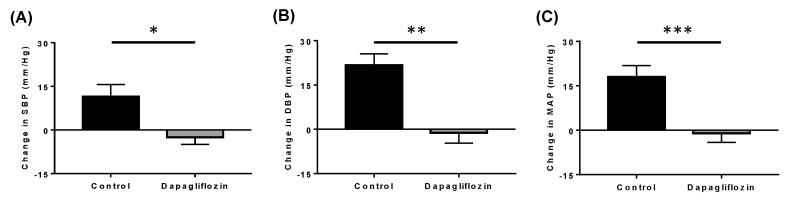
SGLT2 inhibition with DAPA treatment prevents the increase in blood pressure in neurogenically hypertensive mice: Effects of DAPA treatment on (**A**) systolic blood pressure, (**B**) diastolic blood pressure and (**C**) mean arterial blood pressure were measured using a tail-cuff apparatus. *n* = 10–12 mice/group; * *p* = 0.006; ** *p* = 0.0003; *** *p* = 0.0008; All data represented as mean ± SEM. Figure taken from [22].

**Figure 5 ijms-24-14243-f005:**
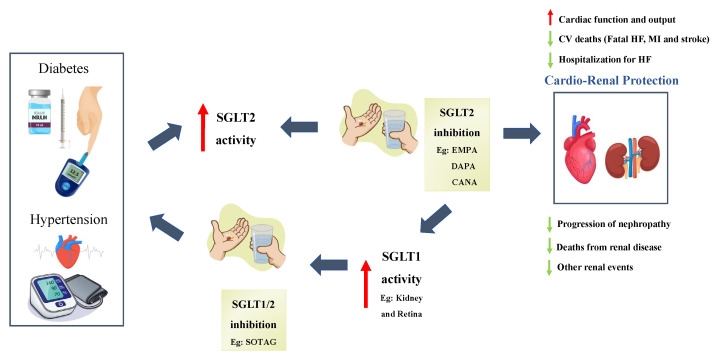
Cardiorenal benefits of Sodium–Glucose Cotransporter 2 (SGLT2) inhibitors and proposed effects on SGLT1 activity: Sodium–Glucose Cotransporter 2, SGLT2; Sodium–Glucose Cotransporter 1, SGLT1; Empagliflozin, EMPA; Dapagliflozin, DAPA; Canagliflozin, CANA; CV, cardiovascular; HF, heart failure and MI, myocardial infarction. SGLT1/2 Inhibition: [83,97], SGLT1 Activity: [85,90], Cardio-Renal Protection: [25,27,51,54,96]. Red arrows indicate an increase in the parameter. Green arrows indicate a decrease in the parameter. Blue arrows indicate the flow of the figure.

**Table 3 ijms-24-14243-t003:** Clinical trials in patients irrespective of diabetic status.

**Clinical Trials in Patients with Diabetes Only**					
**Trial/Year**	**SGLT2 Inhibitor**	**Patient Cohort**	**Patient #**	**Outcome**	**Ref #**
2015 EMPA-REG	Empagliflozin	T2D—High Risk of Cardiovascular Events	7064	↓ Mortality & Hospitalisation due to HF ↓ Risk of Clinically Relevant Renal Events	[25,51]
2017 CANVAS	Canagliflozin	T2D—High Risk of CVD	10,143	↓ Major Adverse Cardiovascular Events ↓ Albuminuria Levels ↓ Renal Replacement Therapy/Death	[27]
2018 DELCARE-TIMI	Dapagliflozin	T2D—High Risk of Atherosclerotic CVD	17,190	↓ Cardiovascular Death or Hospitalisation for HF.	[26]
2018 CREDENCE	Canagliflozin	T2D and Kidney Disease	4401	↓ Risk of Kidney Failure/Cardiovascular Events	[52]
2019 VERTIS	Ertugliflozin	T2D and CVD	8246	↓ First/Total Hospitalization ↓ Risk of death from HF/CVD	[53]
**Clinical Trials in Patients Irrespective of Diabetic Status**					
**Trial/Year**	**SGLT2 Inhibitor**	**Patient Cohort**	**Patient #**	**Outcome**	**Ref #**
2019 DAPA-HF	Dapagliflozin	Class II, III or IV Heart Failure & EF < 40%	4744	↓ Worsening Heart Failure/Death from CV Events	[54]
2019 DIAMOND	Dapagliflozin	Non Diabetic Patients with CKD	50	Induced an acute and reversible decline in mGFR levels.	[55]
2020 EMPEROR-REDUCED	Empagliflozin	Class II, III or IV Heart Failure & EF < 40%	3730	↓ Mortality & Hospitalisation due to HF	[56]
2020 DAPA-CKD	Dapagliflozin	High Risk of Kidney & CDV Outcomes	4304	↓ Risk of a declining GFR level ↓ End Stage Renal Disease/Death from Renal Causes	[57]
2021 EMPEROR-PRESERVED	Empagliflozin	Class II, III or IV Heart Failure & EF > 40%	5988	↓ Cardiovascular Death and Hospitalisation for HF.	[58]
2022 EMMY	Empagliflozin	Recent acute myocardial infarction	476	↓ Risk of a declining GFR level ↓ ES Renal Disease/Death from Renal Causes	[59]
2022 EMPA-KIDNEY	Empagliflozin	Patients with CKD	6609	↓ Progression of Kidney Disease ↓ Death from Cardiovascular Causes	[60]
2022 DELIVER	Dapagliflozin	Heart Failure, Left Ventricular EF > 40%	6263	↓ Worsening Heart Failure/Death from Cardiovascular Events	[61]

Arrows represent decreases of parameters in the table. # represents the number of patients.

## Data Availability

The data presented in this study are available on request from the corresponding author.

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
