# Peer review of "The Impact of SGLT2 Inhibitors in the Heart and Kidneys Regardless of Diabetes Status"

_ijms, 2023, doi:10.3390/ijms241814243_

Round 1
Reviewer 1 Report
In this comprehensive review, Matthews et al. summarized the cardiorenal effect of SGLT2i in patients with and without diabetes.
Major:
The review seems to be focusing on cardiorenal effect of SGLT2i, regardless of diabetes status. Title should be changed to emphasize heart and kidney.
The cardiorenal protective effects of SGLT2i are pluripotent and not entirely elucidated. I would suggest authors rephrase the last sentence of abstract to reflect this.
Section 1 Introduction needs to be reorganized to better describe the background of CKD, CVD, DM. Authors also need to distinguish between CKD and ESKD.
Last paragraph of Section 1 fails to mention a promising drug in this space, i.e., novel MR Antagonist such as Finerenone.
Section 2 last paragraph which described GFR and SGLT2i needs to be revised. This is an active area of research, and GFR cut-off for SGLT2i initiation is being modified constantly, now down to 20. Furthermore, there is trials evaluating SGLT2i in dialysis patients.
Section 3.1/4.1 did not fully summarize the known effects of SGLT2i, which also includes erythropoiesis and uric acid reduction etc.
Section 3 and 4 should provide a figure with timeline of various SGLT2i trials and coverage of disease being studied.
Minor:
In general, I would avoid using terms like diabetics; instead, it’s better to say patients or people with diabetes.
overall good but needs some minor improvement in a few parts
Reviewer 2 Report
- I suggest mentioning and briefly summarizing (perhaps as a figure) traditional atherosclerosis and non-traditional atherosclerosis risk factors that culminate in CKD patients and lead to increased cardiovascular morbidity and mortality.
- You should add finerenone as a novel treatment in the introduction section as well, as it is an important drug currently used for treating DKD with albuminuria.
- Line 110-115: this is very controversial, especially because SGLT-2 inhibitors have several other indications as well. They are not registered for treating DM in CKD patients, but can be used in CKD patients due to their nephro- and cardioprotective effects, up until starting dialysis. Some authors also argue that their use is safe in HD/PD patients, especially if they have additional heart failure. Therapeutic nihilism is a problem in this population and they should not be forgotten.
Round 2
Reviewer 1 Report
satisfied with revision